# Charlson–Deyo Comorbidity Index as a Novel Predictor for Recurrence in Non-Muscle-Invasive Bladder Cancer

**DOI:** 10.3390/cancers15245770

**Published:** 2023-12-08

**Authors:** Lukas Scheipner, Hanna Zurl, Julia V. Altziebler, Georg P. Pichler, Stephanie Schöpfer-Schwab, Samra Jasarevic, Michael Gaisl, Klara C. Pohl, Karl Pemberger, Stefan Andlar, Georg C. Hutterer, Uros Bele, Conrad Leitsmann, Marianne Leitsmann, Herbert Augustin, Richard Zigeuner, Sascha Ahyai, Johannes Mischinger

**Affiliations:** 1Department of Urology, Medical University of Graz, 8010 Graz, Austria; hanna.zurl@medunigraz.at (H.Z.); juliavalerie.altziebler@uniklinikum.kages.at (J.V.A.); georg.pichler@medunigraz.at (G.P.P.); stephanie.schoepfer@medunigraz.at (S.S.-S.); samra.jasarevic@medunigraz.at (S.J.); michael.gaisl@uniklinikum.kages.at (M.G.); klara.pohl@medunigraz.at (K.C.P.); k.pemberger@medunigraz.at (K.P.); stefan.andlar@stud.medunigraz.at (S.A.); georg.hutterer@medunigraz.at (G.C.H.); uros.bele@uniklinikum.kages.at (U.B.); conrad.leitsmann@medunigraz.at (C.L.); marianne.leitsmann@medunigraz.at (M.L.); herbert.augustin@medunigraz.at (H.A.); richard.zigeuner@medunigraz.at (R.Z.); sascha.ahyai@medunigraz.at (S.A.); johannes.mischinger@uniklinikum.kages.at (J.M.); 2Institute for Applied Quality Improvement and Research in Health Care, 37073 Goettingen, Germany

**Keywords:** CCI, NMIBC, recurrence, Charlson comorbidity index, predictor

## Abstract

**Simple Summary:**

For certain malignancies, comorbidities are associated with an increased risk of cancer recurrence. However, it is unknown if this is also true for localized, non-muscle-invasive bladder cancer. The Charlson–Deyo comorbidity score is a commonly used tool for comorbidity-associated survival. We relied on this tool to stratify patients into low vs. high comorbidity burden groups and tested whether the comorbidity burden affected the risk of recurrence. Our data suggest that patients who harbor a high comorbidity burden have an increased risk of tumor recurrence.

**Abstract:**

Purpose: To test the association between the Charlson–Deyo Comorbidity Index (CCI) and the recurrence of non-muscle-invasive bladder cancer (NMIBC). Methods: NMIBC (Ta, T1, TIS) patients who underwent transurethral resection of bladder tumor (TURB) between 2010 and 2018 were identified within a retrospective data repository of a large university hospital. Kaplan–Meier estimates and uni- and multivariable Cox regression models tested for differences in risk of recurrence according to low vs. high comorbidity burden (CCI ≤ 4 vs. >4) and continuously coded CCI. Results: A total of 1072 NMIBC patients were identified. The median follow-up time of the study population was 55 months (IQR 29.6–79.0). Of all 1072 NMIBC patients, 423 (39%) harbored a low comorbidity burden vs. 649 (61%) with a high comorbidity burden. Overall, the rate of recurrence was 10% at the 12-month follow-up vs. 22% at the 72-month follow-up. In low vs. high comorbidity burden groups, rates of recurrence were 6 vs. 12% at 12 months and 18 vs. 25% at 72 months of follow-up (*p* = 0.02). After multivariable adjustment, a high comorbidity burden (CCI > 4) independently predicted a higher risk of recurrence (HR 1.42, 95% confidence interval (CI) 1.06–1.92, *p* = 0.018). After multivariable adjustment, the hazard of recurrence increased by 5% per each one-unit increase on the CCI scale (HR 1.05, 95% CI 1.00–1.10, *p* = 0.04). Conclusions: Comorbidities in NMIBC patients are common. Our data suggest that patients with higher CCI have an increased risk of BC recurrence. As a consequence, patients with a high comorbidity burden should be particularly encouraged to adhere to NMIBC guidelines and conform to follow-up protocols.

## 1. Introduction

With nearly 600,000 new diagnoses each year, bladder cancer (BC) ranks as the tenth most common form of cancer worldwide [1]. Its incidence is notably higher in men, approximately four times more than in women [1]. BC is predominantly diagnosed in the non-muscle-invasive (NMIBC) stage, where transurethral resection of the bladder tumor (TURB) serves as the standard of care. Despite the generally favorable prognosis, tumor recurrence occurs in 30–80% of all patients [2]. Moreover, the diagnostic follow-up and subsequent treatment of NMIBC have a substantial impact on healthcare systems [3]. Due to the high recurrence rate and the corresponding follow-up protocols, BC incurs the highest lifetime treatment costs per patient among all malignant diseases [4]. The consequential impact on healthcare systems, both in terms of diagnostic follow-up and treatment costs, emphasizes the urgency for advancements in BC management. In order to enhance patient outcomes and optimize resource utilization, more precise, risk-related tools and strategies are needed [3].

Several factors, such as age, number of tumors, size, stage, grade, and concomitant carcinoma in situ (CIS), have previously been identified as affecting the risk of recurrence [5,6,7]. However, it is currently unknown if patients’ comorbidities also influence recurrence rates in NMIBC patients. The relationship between comorbidities and cancer-specific survival outcomes in BC patients has already been extensively studied. However, its effect on tumor biology, progress, metastatic spread, and recurrence in NMIBC are still largely unknown [8,9,10,11]. In addition, several previous studies have linked certain comorbidities, such as diabetes, obesity, senescence, or cardiovascular disease, to a higher risk of recurrence in other malignancies [12,13]. 

The Charlson–Deyo Comorbidity Index (CCI) is one of the most widely used and validated scoring systems for comorbidity-associated survival and is frequently assessed during pre-operative evaluations [14]. The CCI represents an adaption of the original Charlson Comorbidity Index that is better suited for contemporary diagnosis codes [15]. Each condition is assigned a score (1, 2, 3, or 6), and these scores are then summed to provide a final CCI score that ranges from 0 (no comorbidities) to 17 (highest comorbidity burden). Given the inherent simplicity and broad accessibility of the CCI, it holds promise as an effective tool for further stratifying the risk of recurrence among NMIBC patients.

We hypothesized that NMIBC patients with a high comorbidity burden have an elevated risk of BC recurrence. We further hypothesized that with each increase in absolute CCI score, there is a corresponding increase in the hazard of BC recurrence. We relied on a large, single-institution dataset of NMIBC patients to test these hypotheses.

## 2. Methods

### 2.1. Study Population

The study was conducted at the Department of Urology, Medical University of Graz (MUG). After obtaining institutional review board approval (EK: 31-228 ex18/19), we retrospectively analyzed the data of 1072 NMIBC patients who underwent TURB between 2010 and 2018. Written informed consent was not obtained from individual patients because the local ethics committee specifically granted a “waiver of consent” for this retrospective database study. All investigations were in accordance with the principles embodied in the Declaration of Helsinki.

After primary TURB, a second resection was performed within 6 weeks if the criteria according to the European Association of Urology (EAU) guidelines at the time of surgery were met [16]. Subsequently, patients were followed-up with during clinical check-ups and regular cystoscopies, which were performed in the outpatient clinic of the Department of Urology of the MUG or by outpatient urologists. If a potential BC recurrence was identified, patients were rescheduled to our clinic for subsequent TURB. CCI was retrospectively assessed based on the patient’s medical history and comorbidities.

### 2.2. Variables of Interest and Outcomes

Since the first introduction of the CCI, several adaptions have been proposed [17]. For the purpose of our study, we relied on the revised Charlson–Deyo Comorbidity Index, as it is well-established and better suited for contemporary ICD codes. The Charlson–Deyo Comorbidity Index is calculated by assigning a weighted score to each comorbid condition based on its severity. The scores are then summed to obtain a total score for a given patient. The weights are derived from the original Charlson Comorbidity Index, and as mentioned in the revised version, these weights were adapted to align with contemporary International Classification of Diseases (ICD) codes. The comorbid conditions considered in the index are assigned scores ranging from 1 to 6, with higher scores indicating greater severity and a potentially higher impact on overall health. The total score provides an estimate of the patient’s overall comorbidity burden, with a higher score corresponding to a higher predicted risk of mortality [15]. 

Demographic covariates consisted of age (continuously coded), sex (male vs. female), Body Mass Index (BMI; continuously coded), and smoking status (no smoker vs. smoker vs. unknown). Tumor characteristics (at first TURB) consisted of stage (pTa low-grade (LG) vs. pTa high-grade (HG) vs. pT1 vs. pTis), EAU risk groups (version 2020; low vs. intermediate vs. high vs. very high) [16], concomitant carcinoma in situ (CIS), presence of the detrusor muscle in histology, and adjuvant therapy (either immediate postoperative intravesical chemotherapy or Bacillus Calmette–Guérin (BCG)). The covariates were chosen based on their recognized associations with the outcomes of NMIBC patients [16,18,19,20,21,22]. The endpoint of the study was a recurrence of primary NMIBC.

### 2.3. Statistical Analysis

Descriptive statistics included frequencies and proportions for categorical variables. Medians and interquartile ranges (IQR) were reported for continuously coded variables. Kruskal–Wallis and Chi-square tests were used to assess differences in medians and proportions. The follow-up period for each patient was defined as the time from the first TURB to the date of the last check-up or, in the case of recurrence, to the date of a subsequent TURB.

We relied on the following steps to test the association between CCI and the risk of recurrence: 

First, Kaplan–Meier curves depicted a recurrence over time according to low vs. high comorbidity burden [23]. A cut-off of CCI > 4 was chosen for high comorbidity burden based on the median CCI of the cohort. 

Second, univariable and multivariable Cox regression models tested for differences in the risk of recurrence according to low vs. high comorbidity burden [24]. Here, adjustment variables for the multivariable model consisted of sex, stage, concomitant CIS, adjuvant therapy, and histological detrusor muscle presence.

Third, univariable and multivariable Cox regression models tested for the association between continuously coded CCI and the risk of recurrence. Similarly, adjustment variables for the multivariable model consisted of sex, stage, concomitant CIS, adjuvant therapy, and histological detrusor muscle presence.

In all statistical analyses, an R software environment for statistical computing and graphics (R version 4.1.2; R Foundation for Statistical Computing, Vienna, Austria) was used. All tests were two-sided, with a level of significance set at *p* < 0.05. All statistical analyses, as well as all the included tables and figures, aimed to follow the principles of the current guidelines for reporting statistics in urologic research [25,26].

## 3. Results

### 3.1. Descriptive Characteristics

We identified a total of 1072 NMIBC patients who underwent TURB at our department between 2010 and 2018. The median age of our study population was 71 years. Of all of the patients, 256 (24%) were female. The median CCI was 4 (IQR 3–4.5). Of all of the patients, 566 (52%) had pTa LG disease vs. 94 (9%) pTa HG vs. 295 (28%) pT1 vs. 29 (3%) pTis vs. 88 (8%) unknown stage. A total of 73 (6.8%) patients had concomitant CIS. Of all of the patients, 63 (6%) were EAU low vs. 471 (44%) intermediate vs. 484 (45%) high vs. 48 (5%) very high risk. The detrusor muscle was present in 47% (507) of all histologic samples. Overall, 52% (554) received postoperative intravesical chemotherapy, and 19% (206) received subsequent BCG therapy. Statistically significant differences between the low vs. high comorbidity burden group existed for age (62 vs. 77 years, *p* < 0.001), length of surgery (24 vs. 29 min, *p* = 0.002), smoking status (smoker 39 vs. 27%, *p* < 0.001), stage (e.g., pT1 23 vs. 32%, *p* < 0.001), EAU risk classification (e.g., high 36 vs. 51%, *p* < 0.001), and postoperative chemotherapy (received 58 vs. 47%, *p* < 0.001; Table 1).

### 3.2. Overall Recurrence of the Study Population

The median follow-up time of the study population was 55 months (IQR 29.6–79.0). Over the course of the study period, a total of 210 patients had tumor recurrence. At the 12-month follow-up, the rate of recurrence was 10%. At the 72-month follow-up, the rate of recurrence was 22%. The median time to recurrence was not reached.

### 3.3. Risk of Recurrence According to Low vs. High Comorbidity Burden

Of all 1072 NMIBC patients, 423 (39%) harbored a low comorbidity burden vs. 649 (61%) with a high comorbidity burden. The rates of recurrence for low vs. high comorbidity burden groups were 6 vs. 12% at the 12-month follow-up and 18 vs. 25% at the 72-month follow-up (*p* = 0.02; Figure 1). In the univariable Cox regression analysis, a high comorbidity burden was associated with a higher risk of recurrence (HR 1.40, 95% CI 1.05–1.87, *p* = 0.02). After multivariable adjustment, a high comorbidity burden independently predicted a higher risk of recurrence (HR 1.42, 95% CI 1.06–1.92, *p* = 0.018; Table 2).

### 3.4. Risk of Recurrence According to Charlson–Deyo Comorbidity Index as a Continuous Scale

In the univariable Cox regression analysis, CCI was associated with a higher risk of recurrence (HR 1.05, 95% CI 1.00–1.10, *p* = 0.03). After the multivariable adjustment, the hazard of recurrence increased by 5% per each one-unit increase on the CCI scale (HR 1.05, 95% CI 1.00–1.10, *p* = 0.04; Table 3).

## 4. Discussion

The relationship between comorbidities and the risk of recurrence in NMIBC patients is currently unknown. However, several previous studies have linked certain comorbidities, such as diabetes, obesity, senescence, or cardiovascular disease, to a higher risk of recurrence in other malignancies [12,13]. Based on previous publications in which several specific comorbidities were identified as risk factors for cancer recurrence, we hypothesized that a higher comorbidity burden, based on the CCI, is also associated with an increased risk of BC recurrence [5,6]. Moreover, we hypothesized that with each increase in CCI, there is a corresponding increase in the hazard of BC recurrence. Our analysis resulted in several noteworthy observations.

First, we identified 1072 non-muscle-invasive bladder cancer (NMIBC) patients who underwent transurethral resection of bladder (TURB) at a prominent tertiary care center between 2010 and 2018. Despite relying on a single-institutional database, the substantial size of our NMIBC patient cohort is comparable to other expansive multi-institutional datasets. For instance, Matulewicz et al. scrutinized 723 NMIBC patients from a multi-institutional database, investigating the association of smoking status with the risk of recurrence [27].

Similarly, Miyake et al. relied on 1490 NMIBC patients identified within a multi-institutional dataset to propose a novel risk stratification model for intravesical tumor recurrence [28]. Although recurrence in BC is frequent, a large study population is crucial in order to apply robust statistical testing and reduce the risk of potential bias when testing for potential risk factors. Moreover, a substantial cohort ensures the statistical power needed to detect meaningful associations, enhancing both the reliability and validity of the study. In the context of investigating cancer recurrence, a sizable dataset provides a more comprehensive representation of the population, allowing for more accurate generalizations and reducing the impact of chance variations. 

Second, we recorded important differences in baseline characteristics in low vs. high comorbidity burden patients. Specifically, patients with a high comorbidity burden (CCI > 4) had higher proportions of pT1 stage (32 vs. 21%, *p* < 0.001) and EAU high-risk classification (51 vs. 36%, *p* < 0.001). Moreover, they were less likely to receive postoperative chemotherapy (47 vs. 58%, *p* < 0.001). These factors are strongly associated with a higher risk of recurrence. Conversely, no differences between the two groups existed for concomitant CIS (6.9 vs. 6.6%, *p* = 0.8) and adjuvant BCG therapy (19 vs. 19%, *p* > 0.9). Interestingly, patients with higher CCI had a lower proportion of smokers (27 vs. 39%, *p* < 0.001); however, this information was unfortunately unavailable for the majority of patients and therefore inconclusive (57%).

Third, we tested the association between CCI and the risk of tumor recurrence in NMIBC patients. Specifically, we relied on two methodological approaches. First, we identified and subsequently compared patients with low vs. high comorbidity burden based on the median CCI in our patient population (CCI = 4). Here, patients with a high comorbidity burden had a higher risk of recurrence (HR 1.40, 95% CI 1.05–1.87, *p* = 0.02). After multivariable adjustment, a high comorbidity burden remained an independent predictor of recurrence (HR 1.42, 95% CI 1.06–1.92, *p* = 0.018). Second, we addressed CCI on a continuous scale. Here, after multivariable adjustment, the hazard of recurrence increased by 5% per each unit increase on the CCI scale (HR 1.05, 95% CI 1.00–1.10, *p* = 0.04).

Although our data clearly suggest a link between elevated CCI and an increased risk of recurrence, we can only hypothesize about its nature. Considering that BC is predominantly diagnosed in older individuals, it is not unusual for patients in this age group to have a substantial comorbidity burden. This observation is reflected by the relatively high median CCI in our study cohort. In a recent article by Panigrahi et al., the authors discussed current biological evidence that links comorbidities to cancer development and adverse tumor biology [8]. The authors argued that although three-quarters of all cancer patients suffer from at least one comorbidity, its effect on cancer outcomes is still not fully understood. Available evidence suggests that comorbidities exert their effects on tumor biology by altering the tumor microenvironment, cancer metabolism, gut microflora, as well as the response to several cancer therapies [8]. For example, obesity and diabetes are cancer risk factors, as well as frequent comorbidities that can lead to changes to the gut microbiome. Imbalances in the gut microbiome increase the risk of mucosal infections, which subsequently increase the risk of cancer and cancer recurrence [8]. Similarly, chronic infections can cause persistent inflammation, which is a risk factor for cancer development. However, according to the authors, many aspects of the inflammation-to-cancer axis remain poorly understood for most malignancies [8]. Moreover, the role of inflammation in comorbidity-induced cancer progression and recurrence has received limited attention. Although significant progress has been made in the understanding of the underlying biological correlations, Panigrahi et al. conclude that many questions regarding the effect of comorbidities on cancer outcomes are still unanswered. Despite these unknowns, our findings may be of clinical importance. A timely detection of BC recurrence is crucial to reducing the risk of progression. This holds particular importance for patients with a high comorbidity burden, as they often face limited suitability for a cystectomy. As such, patients with a high comorbidity burden should be strongly encouraged to adhere to NMIBC guidelines and conform to follow-up protocols. To the best of our knowledge, we are the first to report an association between CCI and the risk of recurrence in NMIBC patients. Unfortunately, in consequence, our results cannot be compared with other previous reports. Future studies relying on similarly large or bigger databases should ideally validate or refute the current study’s findings. With this in mind, it becomes apparent that a better understanding of the relationship between comorbidities and BC recurrence is crucial for further improving the prognostic assessments of NMIBC patients. If validated in other similarly large cohorts of NMIBC patients, CCI could emerge as a valuable risk stratification tool in the management of NMIBC patients. The identification of a higher comorbidity burden as a potential risk factor for recurrence may have implications for risk stratification and tailored management strategies in NMIBC patients. As such, we believe that further exploration of this association holds promise for refining the prognostic assessments and further advancing personalized treatment and follow-up approaches in NMIBC patients.

Altogether, our comprehensive analyses shed light on the previously unexplored relationship between comorbidities, measured by the CCI, and the risk of recurrence in NMIBC patients. Our findings indicate that a significant link between higher CCI and increased risk of recurrence exists. As we expected, comorbidities in BC patients are common and can cause significant limitations in the management of these patients. Our results emphasize the clinical relevance of vigilant follow-up protocols, especially for patients with an elevated comorbidity burden. Although the exact biological mechanisms behind this association remain a subject of ongoing investigation, its potential clinical impact seems promising. If validated by other similarly large studies, the CCI could emerge as a valuable tool for risk stratification in the treatment and follow-up strategies in NMIBC patients.

Despite the novelty of our findings, our study is not devoid of limitations that necessitate acknowledgment. Firstly, while our analyses revealed a statistically significant correlation between CCI and the risk of recurrence in NMIBC patients, it is imperative to clarify that this association does not inherently establish a causal relationship. Although biological correlation evidence exists, the full understanding of how comorbidities affect cancer recurrence remains incomplete. Nevertheless, we employed multivariable analyses to minimize potential confounders. Second, the relatively lower event rate in our cohort, compared to other study groups, could be attributed to the median time to follow-up not reaching maturity [29,30,31]. Despite this, the observed difference in recurrence rates is evident at the 12-month follow-up mark. Third, while substantial in scale, our data are confined to a single institution, potentially introducing selection bias. Consequently, we urge other study groups to explore the impact of CCI or alternative comorbidity markers on NMIBC recurrence risk for comprehensive validation. Despite these limitations, our study provides valuable insights into the association between comorbidities and risk of recurrence in NMIBC. Nevertheless, these limitations underscore the importance of further investigations and highlight the necessity for collaborative efforts across diverse clinical settings to validate or refute our results.

## 5. Conclusions

Comorbidities in patients undergoing TURB for NMIBC are frequent and often shape the subsequent management strategies. Our data suggest that patients with higher CCI have an increased risk of BC recurrence. To the best of our knowledge, we are the first to report this association in NMIBC patients. In consequence, patients with a high comorbidity burden should be particularly encouraged to adhere to guidelines and conform to follow-up protocols.

The significance of early detection in these cases is underscored by the fact that patients with higher comorbidity burdens often face limited suitability for more radical interventions, namely cystectomies. Therefore, the emphasis on stringent follow-up protocols becomes a critical component in the clinical management of this patient population in order to achieve favorable outcomes.

Although other studies are needed to validate our findings, the CCI holds promise as a valuable tool in further advancing risk stratification in NMIBC patients.

## Figures and Tables

**Figure 1 cancers-15-05770-f001:**
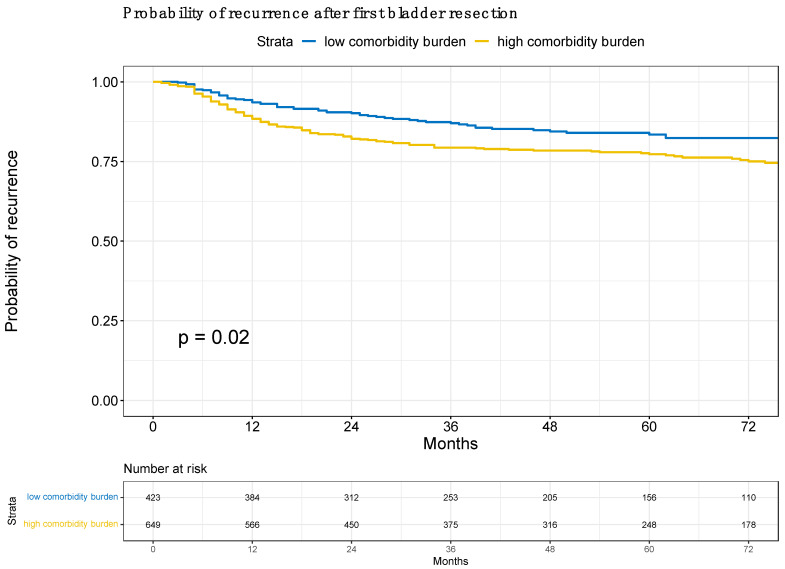
Kaplan–Meier curves depicting the probability of recurrence in non-muscle-invasive bladder cancer (NMIBC) patients, according to low vs. high comorbidity burden, defined by the Charlson–Deyo Comorbidity Index (CCI; low ≤ 4, high > 4).

**Table 1 cancers-15-05770-t001:** Descriptive characteristics of 1072 non-muscle-invasive bladder cancer (NMIBC) patients according to low vs. high comorbidity burden defined by the Charlson–Deyo Comorbidity Index (CCI; low ≤ 4, high > 4).

Characteristic	*N*	CCI Low, N = 423 (39%) ^1^	CCI High, N = 649 (61%) ^1^	*p*-Value ^2^
Age	1072	62 (54, 68)	77 (71, 83)	<0.001
CCI	1072	2.00 (1.00, 3.00)	6.00 (4.00, 7.00)	<0.001
Female sex	1072	108 (26%)	148 (23%)	0.3
BMI	1022	27.1 (24.5, 29.6)	26.7 (24.2, 29.3)	0.2
Smoking status	1072			<0.001
Yes		166 (39%)	173 (27%)	
No		52 (12%)	70 (11%)	
Unknown		205 (48%)	406 (63%)	
Stage	1072			<0.001
pTa low		263 (62%)	303 (47%)	
pTa high		35 (8.3%)	59 (9.1%)	
pT1		89 (21%)	206 (32%)	
pTis		12 (2.8%)	17 (2.6%)	
Unknown		24 (5.7%)	64 (9.9%)	
EAU risk classification	1072			<0.001
low		38 (9.0%)	25 (3.9%)	
intermediate		212 (50%)	259 (40%)	
high		153 (36%)	331 (51%)	
very high		18 (4.3%)	30 (4.6%)	
Unknown		2 (0.5%)	4 (0.6%)	
Concomitant CIS	1072			0.8
yes		28 (6.6%)	45 (6.9%)	
no		387 (91%)	595 (92%)	
Unknown		8 (1.9%)	9 (1.4%)	
Detrusor muscle in histology	1072			0.9
yes		200 (47%)	307 (47%)	
no		144 (34%)	213 (33%)	
Unknown		79 (19%)	129 (20%)	
Postoperative chemotherapy	1072			<0.001
yes		246 (58%)	308 (47%)	
No		159 (38%)	320 (49%)	
Unknown		18 (4.3%)	21 (3.2%)	
Adjuvant BCG	1072	81 (19%)	125 (19%)	>0.9

^1^ Median (IQR); n (%). ^2^ Wilcoxon rank sum test; Pearson’s Chi-square test; Fisher’s exact test. Abbrev.: Charlson–Deyo Comorbidity Index (CCI); Body Mass Index (BMI); European Association of Urology (EAU); Carcinoma in situ (CIS); Bacillus Calmette–Guérin (BCG).

**Table 2 cancers-15-05770-t002:** Univariable and multivariable logistic regression models predicting risk of recurrence in non-muscle-invasive bladder cancer (NMIBC) patients according to comorbidity burden (low vs. high) defined by the Charlson–Deyo Comorbidity Index (CCI, low ≤ 4, high >4).

	Univariable			Multivariable ^2^		
Characteristic	OR ^1^	95% CI ^1^	*p*-Value	OR ^1^	95% CI ^1^	*p*-Value
Comorbidity burden						
low	—	—		—	—	
high	1.40	1.05, 1.87	0.02	1.42	1.06, 1.92	0.018

^1^ OR = odds ratio; CI = confidence interval. ^2^ Adjusted for sex, stage, concomitant CIS, adjuvant therapy, and histological detrusor muscle presence.

**Table 3 cancers-15-05770-t003:** Univariable and multivariable logistic regression models predicting risk of recurrence in non-muscle-invasive bladder cancer (NMIBC) patients according to the Charlson–Deyo Comorbidity Index (CCI).

	Univariable			Multivariable ^2^		
Characteristic	OR ^1^	95% CI ^1^	*p*-Value	OR ^1^	95% CI ^1^	*p*-Value
CCI	1.05	1.00, 1.10	0.03	1.05	1.00, 1.10	0.04

^1^ OR = odds ratio, CI = confidence interval. ^2^ Adjusted for sex, stage, concomitant CIS, adjuvant therapy, and histological detrusor muscle presence.

## Data Availability

The code for the analyses will be made available upon request.

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
