# Peer review of "Charlson–Deyo Comorbidity Index as a Novel Predictor for Recurrence in Non-Muscle-Invasive Bladder Cancer"

_cancers, 2023, doi:10.3390/cancers15245770_

Round 1
Reviewer 1 Report
Comments and Suggestions for Authors
The authors should be congratulated for their great work. The topic addressed is one of the major concerns of NMIBC. The recurrence accounts for the vast majority of intermediate-to-high-grade patients. However, the role of comorbidities in predicting recurrence is a novel topic worthy of more attention. The authors described properly the backgrounds. Their results, moreover, are worthy of publication, achieving a goal that could edit the follow-up schedule of NMIBC patients based on their comorbidities.
Author Response
We thank the Reviewers for these kind words and their time to review our manuscript!
Reviewer 2 Report
Comments and Suggestions for Authors
This study aimed to assess the association between the Charlson-Deyo Comorbidity Index (CCI) and recurrence in non-muscle-invasive bladder cancer (NMIBC) patients. The research involved 1,072 NMIBC patients who underwent transurethral resection of bladder tumors (TURB) between 2010 and 2018. Findings revealed that patients with higher CCI had a significantly increased risk of bladder cancer recurrence, emphasizing the importance of close adherence to NMIBC guideline-based follow-up for those with a high comorbidity burden.
Comments for authors:
Title:
The title accurately reflects the content of the study and is concise and informative.
Abstract:
1. Even with the restrictions of the abstract length. While the section is generally well-structured and clear, it could benefit from a more detailed description of the data collection process, including patient selection criteria and potential sources of bias
Introduction:
1. While the introduction is informative, it could be further improved by providing more context on the Charlson-Deyo Comorbidity Index (CCI). Briefly explaining what CCI is and how it is calculated would help readers who may not be familiar with this index.
2. The introduction could benefit from a more in-depth explanation of the significance of NMIBC recurrence in terms of patient outcomes and resource utilization. This would help the reader appreciate the broader implications of the research.
3. To enhance the engagement of the readers, the introduction could incorporate a brief statement about the potential real-world impact of the study findings or the significance for clinicians and patients.
Methods:
1. While the section provides an overview of demographic covariates, tumor characteristics, and endpoint of the study, it could be enhanced by briefly explaining the clinical significance of some variables. For instance, why are age, BMI, and smoking status important in this context? Providing a brief rationale would be beneficial for readers.
2. The statistical analysis description is detailed but could be simplified for better readability. It may be beneficial to use bullet points or subheadings to clearly delineate the different steps of the analysis.
3. The description mentions the use of Kaplan-Meier curves and Cox regression models, but it could benefit from a brief statement about the choice of these statistical methods and why they are appropriate for this study.
4. There is a reference to "current guidelines for reporting statistics in urologic research," which is positive. However, it would be helpful to include a citation to these guidelines for readers who may want to refer to them for more details.
Results:
1. While the tabular format (Table 1) is informative, it could be enhanced by adding brief explanations of the clinical significance of the variables presented. This would help readers who may not be familiar with the clinical context.
2. The description of the risk of recurrence based on comorbidity burden could benefit from a more detailed interpretation of the findings. For instance, discussing the clinical implications of a 6% difference in recurrence rates at 12 months and the impact of a 5% increase in the hazard of recurrence per unit increase in CCI scale would make the results more meaningful to the readers.
3. Including confidence intervals for the reported hazard ratios would improve the interpretation of the statistical significance of the findings.
Discussion:
1. While the section provides strong hypotheses, it could benefit from a brief statement about the potential clinical implications of the research. For example, how would understanding the relationship between comorbidities and BC recurrence impact patient care and treatment decisions?
2. The reference to Panigrahi et al.'s article about the biological evidence linking comorbidities to cancer development and tumor biology is valuable. However, the section could be enhanced by briefly summarizing the key points from that article to provide readers with more context.
Conclusions:
1. The statement suggests that the research has practical implications for clinical practice, emphasizing the importance of follow-up protocols for high comorbidity burden patients. However, it could be further enhanced by briefly mentioning the potential benefits of early detection of BC recurrence and how it may influence treatment decisions and patient outcomes.
Author Response
Response to the Reviewers
Reviewer 2:
Title:
The title accurately reflects the content of the study and is concise and informative.
A: We thank the Reviewer for this kind comment!
Abstract:
- Even with the restrictions of the abstract length.While the section is generally well-structured and clear, it could benefit from a more detailed description of the data collection process, including patient selection criteria and potential sources of bias
A: We thank the Reviewer for this important comment. We fully agree with the Reviewer that a well -structured abstract is important for the reader. Unfortunately, the author guidelines for this journal are rather strict with the word limitation. Since we already exceed the journals recommended word limit of around 200 words (254 word) we were unfortunately not able to expand the abstract further. We hope that the Reviewer understands our effort to comply with the journal´s recommendations.
Introduction:
- While the introduction is informative, it could be further improved by providing more context on the Charlson-Deyo Comorbidity Index (CCI). Briefly explaining what CCI is and how it is calculated would help readers who may not be familiar with this index.
A: We thank the Reviewer for this comment. We fully agree with the Reviewer that the introduction would benefit from an extended explanation of the CCI. To address this, we have now added the following lines to the introduction:
“The CCI represents an adaption of the original Charlson-Comorbidity Index that is better suited for contemporary diagnoses codes. [11] Each condition is assigned a score (1,2,3 or 6), which are then summed to provide a final CCI score that ranges from 0 (no comorbidities) to 17 (highest comorbidity burden).”
We hope that these changes are in accordance with the Reviewers suggestion.
- The introduction could benefit from a more in-depth explanation of the significance of NMIBC recurrence in terms of patient outcomes and resource utilization. This would help the reader appreciate the broader implications of the research.
A: We thank the Reviewer for this comment. To further emphasise the significance of NMIBC recurrence in terms of patient outcomes and resource utilization we have now added the following in the introduction.
“As a result of the high recurrence rate and the corresponding follow-up protocols, BC has the highest lifetime treatment costs per patient of all malignant disease. [3]”
- To enhance the engagement of the readers, the introduction could incorporate a brief statement about the potential real-world impact of the study findings or the significance for clinicians and patients.
A: We thank the Reviewer for this valuable suggestion. We agree that a brief statement would further enhance the engagement of the readers. To address this, we have added the following to the introduction:
“Given the inherent simplicity and broad accessibility of the CCI, it holds promise as an effective tool for further stratifying the risk of recurrence among NMIBC patients.”
We want to thank the Reviewer for his suggestions regarding our manuscript’s introduction. We believe that the added changes greatly improve the final paper!
Methods
- While the section provides an overview of demographic covariates, tumor characteristics, and endpoint of the study, it could be enhanced by briefly explaining the clinical significance of some variables. For instance, why are age, BMI, and smoking status important in this context? Providing a brief rationale would be beneficial for readers.
A: We thank the Reviewer for this comment. Indeed, we have selected these covariates since they have been shown to affect the outcomes of NMIBC patients. We fully agree with the Reviewer that this should be stated, and that the relevant publications should be cited as a rationale. We have now added the following:
“The covariates were chosen based on their recognized associations to affect the outcomes of NMIBC patients. [15–19] “
- The statistical analysis description is detailed but could be simplified for better readability. It may be beneficial to use bullet points or subheadings to clearly delineate the different steps of the analysis.
A: We thank the Reviewer for this valuable suggestion. We fully agree with the Reviewer that good readability is very important, especially in the methods section of a manuscript. We have now used paragraphs to separate each methodological step. We believe that this improves the readability of our methods section, while simultaneously aligning with the journals established style.
- The description mentions the use of Kaplan-Meier curves and Cox regression models, but it could benefit from a brief statement about the choice of these statistical methods and why they are appropriate for this study.
A: We thank the Reviewer for this comment. Both Kaplan-Meier analyses, as well as Cox regression models represent the most well-known statistical tools in oncologic outcomes research. Given the fact that this journal is mostly read by specialists in the field of oncology or other medical specialties we abstained from further expanding the fundamentals of these two statistical tools. However, to accommodate the Reviewers suggestion we cited appropriate key publications for the interested reader.
- There is a reference to "current guidelines for reporting statistics in urologic research," which is positive. However, it would be helpful to include a citation to these guidelines for readers who may want to refer to them for more details.
A: We thank the Reviewer for spotting this error. We have now addressed this, and added the appropriate citations.
Results:
- While the tabular format (Table 1) is informative, it could be enhanced by adding brief explanations of the clinical significance of the variables presented. This would help readers who may not be familiar with the clinical context.
A: We thank the Reviewer for this suggestion. We agree with the Reviewer that our Table 1 is missing the correct explanations of the abbreviations. This could be confusing for readers who are not familiar with the context of our manuscript. We have now addressed this by adding a legend. We hope that this change is consisted with the Reviewers suggestion. The edited Table 1 will be uploaded together with the other corrections made.
- The description of the risk of recurrence based on comorbidity burden could benefit from a more detailed interpretation of the findings. For instance, discussing the clinical implications of a 6% difference in recurrence rates at 12 months and the impact of a 5% increase in the hazard of recurrence per unit increase in CCI scale would make the results more meaningful to the readers.
A: We thank the Reviewer for this comment. We believe that the results section should be reserved for stating the results of the analyses in the most clear und objective manner. In order to avoid subjective discussion in the results section and to adhere to the journals recommendations we did not include further discussion in this section of the manuscript.
- Including confidence intervals for the reported hazard ratios would improve the interpretation of the statistical significance of the findings.
A: We thank the Reviewer for this very important comment. We fully agree with the Reviewer that including the 95% confidence intervals (CI) in the results section is of very high importance. In our original draft, the 95% CI were already included in the results section, as well as the corresponding Tables. However, at certain sections the phrasing “CI” was missing which may have caused the Reviewer to miss it. We have now addressed this issue.
Discussion:
- While the section provides strong hypotheses, it could benefit from a brief statement about the potential clinical implications of the research. For example, how would understanding the relationship between comorbidities and BC recurrence impact patient care and treatment decisions?
A: We thank the Reviewer for this important comment! We tried to highlight the potential clinical implications of our research in the following section of the manuscript:
“Despite these unknowns, our findings may be of clinical importance. A timely detection of BC recurrence is crucial to reduce to risk of progression. This holds particular importance for patients with high comorbidity burden, as they often face limited suitability for cystectomy. As such, patients with high comorbidity burden should be strongly encouraged to adhere to NMIBC guideline conform follow-up protocols.”
However, given that our manuscript is the first to describe an association between a high comorbidity burden and an increased risk of recurrence, no validation studies exist. As such, we believe it is of utmost importance to carefully phrase the potential clinical implications, in order to avoid statements that are not based on sound evidence. In order to address the Reviewers suggestion, we have added the following at the end of this paragraph:
“If validated in other, similarly large cohorts of NMIBC patients, CCI could emerge as a valuable risk stratification tool in the management of NMIBC patients.”
- The reference to Panigrahi et al.'s article about the biological evidence linking comorbidities to cancer development and tumor biology is valuable. However, the section could be enhanced by briefly summarizing the key points from that article to provide readers with more context.
A: We thank the Reviewer for this comment. Indeed, Panigrahi et al´s article is a very valuable contribution to this field of research, and we can highly recommend it to everyone who is interested in this field. The article itself is a summarization of numerous important studies. In our discussion we introduce and summarize their work as follows:
“In a recent article by Panigrahi et al., the authors discuss current biological evidence that links comorbidities to cancer development and adverse tumor biology. [26] Available evidence suggests that comorbidities exert their effects on tumor biology by altering the tumor microenvironment, cancer metabolism, the gut microflora, as well as the response to several cancer therapies.[26]Although significant progress has been made in the understanding of the underlying biological correlations, Panigrahi et al. conclude that many questions regarding the effect of comorbidities on cancer outcomes are still unanswered. “
Given our already lengthy manuscript, we do not believe that expanding further on these points adds to the overall discussion of our manuscript. There are simply too many studies and areas of research discussed by Panigrahi et al, which cannot be described within a few sentences in a meaningful way. Moreover, we also had to adhere to the suggestions of other Reviewers, who suggested other studies that should be discussed.
We hope that the Reviewer finds this agreeable.
Conclusions:
- The statement suggests that the research has practical implications for clinical practice, emphasizing the importance of follow-up protocols for high comorbidity burden patients. However, it could be further enhanced by briefly mentioning the potential benefits of early detection of BC recurrence and how it may influence treatment decisions and patient outcomes.
A: We thank the Reviewer for this comment. We agree that briefly mentioning the potential benefits of early detection can further enhance our manuscripts conclusion. Our altered conclusion now reads as follows:
“Comorbidities in patients undergoing TURB for NMIBC are common. Our data suggests that patients with higher CCI have an increased risk for BC recurrence. In consequence, patients with high comorbidity burden should be particularly encouraged to adhere to guideline conform follow-up protocols. As these patients often face limited suitability for cystectomy, early detection of recurrence is crucial in order to achieve favorable outcomes. “
Taken together, we want to thank the Reviewer for his time and effort to improve our manuscript. We believe that by addressing the points made by the Reviewer, our manuscript has greatly improved. We hope that the Reviewer agrees and finds our corrections suitable.
Reviewer 3 Report
Comments and Suggestions for Authors
The main topic of the article is to study the utility of Charlson-Deyo Comorbidity Index (CCI) in evaluating the risk of recurrency in patient with non-muscle invasive bladder cancer.
Bladder cancer remains the most common malignancy of the urinary tract. If the bladder tumor is identified, in 75% of cases, urothelial bladder cancer confined to the mucosa (NMIBCa—non-muscle invasive disease) is diagnosed. In the remaining 25–30% of patients, BCa has already invaded deeper layers of the bladder wall (MIBCa—muscle-invasive disease) or formed metastases. Despite the generally good prognosis, tumor recurrence occurs in 30-80% of all patients. For this reason, more precise, risk related tools and strategies are needed to enhance patient outcomes. In this study the authors, using retrospective data from the Department of Urology, Medical University of Graz, evaluate the differences in risk of recurrence according to low vs. high comorbidity burden.
The Charlson comorbidity index predicts the one-year mortality for a patient who may have a range of comorbid conditions. A score of zero means that no comorbidities were found; the higher the score, the higher the predicted mortality rate is.
This study demonstrates that it is feasible to implement the CCI in the evaluation of recurrency and prognosis in patients with NMIBCa.
This article is significant in focusing on the importance of stratify risk of recurrency in patients with NMIBCa using different tools. Relying on the results of these index we can better understand which patients have an increased risk for bladder cancer recurrence and which patients and encouraging them to adhere to proper follow-up procedures after surgery.
Unfortunately, the cohort of patient involved in this study is referring only at one single center, and I would recommend adding data from different database from different countries, erasing the center-related hypothetical bias.
I suggest adding the following scientific article links to the bibliography section for a more accurate representation of the references and the general topic of this study:
- 10.3390/jpm13030512
- 10.1016/j.clgc.2021.12.005
- 10.3390/diagnostics12030586
Comments on the Quality of English LanguageMinor editing.
Author Response
The main topic of the article is to study the utility of Charlson-Deyo Comorbidity Index (CCI) in evaluating the risk of recurrency in patient with non-muscle invasive bladder cancer.
Bladder cancer remains the most common malignancy of the urinary tract. If the bladder tumor is identified, in 75% of cases, urothelial bladder cancer confined to the mucosa (NMIBCa—non-muscle invasive disease) is diagnosed. In the remaining 25–30% of patients, BCa has already invaded deeper layers of the bladder wall (MIBCa—muscle-invasive disease) or formed metastases. Despite the generally good prognosis, tumor recurrence occurs in 30-80% of all patients. For this reason, more precise, risk related tools and strategies are needed to enhance patient outcomes. In this study the authors, using retrospective data from the Department of Urology, Medical University of Graz, evaluate the differences in risk of recurrence according to low vs. high comorbidity burden.
The Charlson comorbidity index predicts the one-year mortality for a patient who may have a range of comorbid conditions. A score of zero means that no comorbidities were found; the higher the score, the higher the predicted mortality rate is.
This study demonstrates that it is feasible to implement the CCI in the evaluation of recurrency and prognosis in patients with NMIBCa.
This article is significant in focusing on the importance of stratify risk of recurrency in patients with NMIBCa using different tools. Relying on the results of these index we can better understand which patients have an increased risk for bladder cancer recurrence and which patients and encouraging them to adhere to proper follow-up procedures after surgery.
Unfortunately, the cohort of patient involved in this study is referring only at one single center, and I would recommend adding data from different database from different countries, erasing the center-related hypothetical bias.
I suggest adding the following scientific article links to the bibliography section for a more accurate representation of the references and the general topic of this study:
- 10.3390/jpm13030512
- 10.1016/j.clgc.2021.12.005
- 10.3390/diagnostics12030586
A: We thank the Reviewer for his kind words, as well as his important suggestions. Indeed, we fully agree with the Reviewer that ideally, our data would not be limited to a single institution, but would come from multiple institutions in different countries. We fully acknowledge this limitation and have highlighted it in the final part of the discussion:
“Third, although large in scale, our data is limited to a single institution. This has the inherent potential to introduce selection bias within our study population. In consequence, we encourage other study groups, to investigate the effect of CCI or other comorbidity markers on the risk of recurrence in NMIBC patients.”
Despite this limitation, we confidently believe that our manuscript is valuable as we are the first to show an association between comorbidity burden and risk of recurrence in NMIBC patients. We hope that our manuscript will encourage other colleagues to plan and conduct similar studies that validate or refute our findings.
Finally, the Reviewer suggested several key publications that we missed to reference. We want to thank the Reviewer for these valuable suggestions, as we believe that they represent important contributions in this field of research. We have now included these references in our bibliography section.
We believe that the changes and additions suggested by all of the Reviewers have greatly improved our manuscript. We hope that the Reviewer agrees with this and finds our revised manuscript fit for publication. We want to thank all Reviewers for their time and effort!
Round 2
Reviewer 3 Report
Comments and Suggestions for Authors
Authors answered all comments and suggestions.
Comments on the Quality of English LanguageMinor editing.